# A Survey of Vaccine-Induced Measles IgG Antibody Titer to Verify Temporal Changes in Response to Measles Vaccination in Young Adults

**DOI:** 10.3390/vaccines7030118

**Published:** 2019-09-19

**Authors:** Hiraku Sasaki, Tomoko Fukunaga, Ai Asano, Yoshio Suzuki, Yuko Nakanishi, Junzi Kondo, Hiroki Ishikawa, Nobuto Shibata

**Affiliations:** 1Department of Health Science, Faculty of Health and Sports Science, Juntendo University, 1-1 Hiraga-gakuendai, Inzai, Chiba 270–1695, Japan; ynakani@juntendo.ac.jp (Y.N.); gold-blend20@outlook.jp (J.K.); nshibata@juntendo.ac.jp (N.S.); 2Section of Health Management, Faculty of Health and Sports Science, Juntendo University, 1-1 Hiraga-gakuendai, Inzai, Chiba 270–1695, Japan; tfukunaga@juntendo.ac.jp (T.F.); auemura@juntendo.ac.jp (A.A.); 3Department of Sports Science, Faculty of Health and Sports Science, Juntendo University, 1-1 Hiraga-gakuendai, Inzai, Chiba 270–1695, Japan; yssuzuki@juntendo.ac.jp; 4Department of Microbiology and Immunology, Showa University School of Medicine, 1-5-8 Hatanodai, Tokyo 142–8555, Japan; h_ishikawa@med.showa-u.ac.jp

**Keywords:** measles, IgG antibody titers, vaccination

## Abstract

In Japan, sporadic measles cases increased rapidly in 2019 compared to the past six years. To clarify the persistence of immunity against measles in young adults, this study explored the persistence of immunoglobulin G (IgG) antibody titers against the measles virus in 17- to 24-year-old young participants who reside in the Chiba prefecture of Japan. Measles-specific IgG antibody titers, determined by enzyme immunoassay in serum samples collected from 506 participants, were assessed through statistical analyses. Multivariable regression analysis revealed that the distribution of measles IgG antibody titers was significantly correlated with a medical history of measles (*P* < 0.05), while there was no significant correlation between the number of vaccinations related to measles IgG titers. Furthermore, measles IgG titers tended to decrease, as revealed by the temporal change in IgG titers, during the elapsed period after the last vaccination (*P* = 0.08). These results indicate that periodic vaccination against measles is required to prevent sporadic measles infection in young and older adults.

## 1. Introduction

Measles is an air-bone disease that is propagated via a virus. As measles causes severe complications and even death, vaccination against the measles virus is a protective means to acquire immunity. Japan achieved measles elimination in March 2015 by the enforced administration of two doses of the measles vaccine and the use of a laboratory-based surveillance system based on the guidelines for the prevention of specific infectious diseases [1]. However, more than 300 cases of measles were reported in Japan in May 2019, and this number has increased rapidly compared to that from 2013 to 2018 [2]. Measles cases were reported in 21 of 47 prefectures, and in particular, one-third of measles cases were found in the Osaka prefecture within the 1st to 10th week of 2019 [2]. Receival of the measles vaccination twice during an individual’s lifetime induces acquired immunity against measles in more than 95% of people [3,4]. In Japan’s measles vaccination program, a person born after April 2006 is required to receive two doses of the measles vaccine as a periodic inoculation at 1–2 years and 5–7 years of age [5]. However, prior to the initiation of this program, only one dose of the vaccine was administered after 12 months of age. Hence, the Ministry of Health, Labour, and Welfare recommended two doses of the measles vaccine for people born before April 2006, with the second inoculation administrated within a period of five years (2008–2012, equivalent to 13 and 18 years of age) as a special measure [5]. Within these five years, the rate of vaccination ranged from 77.3% to 94.3% [6]. Recently, the rate of vaccination with two doses of the measles vaccine has been more than 90% in Japan [7]. Except for the people who received two doses of the vaccine at limited periods, almost all young adults born after April 2006 were vaccinated twice before they attained 7 years of age. Measles outbreak was, thus, unexpected in the later childhood years of these individuals due to the implementation of the measles vaccine program. However, there was insufficient knowledge about the protective antibody titers against measles in young adults who received two doses of the measles vaccine within five years of the age limit. 

Currently, measles cases are based on age bracket. In brief, more than 50% of measles cases occurred in 15- to 29-year-old individuals, whereas 12% of cases occurred in individuals aged less than 10 to 14 years [2]. Almost all young adults around 20 years of age were subjected to periodic inoculation, but all of them did not receive the measles vaccination in the same manner. In particular, temporal changes in the development of protective serum immunoglobulin G (IgG) antibody titers against the measles virus may have occurred, in spite of these individuals having received two doses of the vaccine. However, these temporal changes in measles IgG antibody titers in young adults who were subjected to special measures of the measles vaccine are obscure. One of the essential means to prevent measles outbreak is precision diagnosis through the characterization of historical immunity gaps [8]. Further knowledge on temporal immune responses through cross-sectional studies is required to achieve this [9]. Thus, there is a need to elucidate the factors involved in the risk of infection as well as the means of reducing the occurrence of infection.

To elucidate the persistence of immunity against measles in young adults, this study explores the prevalence of IgG antibody titers against the measles virus in 17- to 24-year-old participants residing in the Chiba prefecture and determines the relationship between temporal changes in these antibody titers and the number of vaccinations.

## 2. Materials and Methods 

### 2.1. Study Design

The surveillance was carried out by obtaining samples of sera from 506 young adults between 17 to 24 years of age, who were first-year students in Juntendo University (Sakura Campus, Inzai, Chiba, Japan), to assess the prevalence of specific IgG antibodies against the measles virus. The serum samples were obtained from January 2018 to April 2018. Simultaneously, vaccine history was collected from each individual’s maternity passbook. The studied participants also filled out questionnaires, aimed at obtaining information about the medical history of vaccination against measles and natural measles infection. The collected information was used for the interpretation of the obtained results.

The study protocol was approved by the Ethics Committee of Faculty of Health and Sports Science, Juntendo University (number 30-2) and informed consent was obtained from all participants and their parents before the study.

### 2.2. Laboratory Methods

To determine measles-specific serum IgG antibody titer, the enzyme immunoassay (EIA) was performed in the laboratory of BML, INC. (Tokyo, Japan) using a measles virus immunoglobulin test kit (Measles IgG-EIA manufactured by Denka Seiken Co. Ltd, Tokyo, Japan). For the virus antigen, the Toyoshima strain was used.

### 2.3. Statistical Analysis

To identify the factors that influence measles-specific IgG antibody titer, multivariable regression analysis was performed. Differences in the IgG antibody titers between the participants that had a medical history of measles and those without a medical history were evaluated using unpaired t-tests. To compare differences in IgG antibody titers among the number of vaccinations and temporal characteristics, a one-way analysis of variance (one-way ANOVA) was employed. When there were significant statistical differences, the data were further analyzed using the Bonferroni post-hoc test in order to determine the significance between the groups. Differences were considered significant for *P* values of less than 0.05.

## 3. Results

### 3.1. Survey Participants

During blood collection for determining antibody titers, 18-year-old participants comprised 80% of the total survey participants, and the remaining 20% of participants were ≥ 19 years old. Of the participants who were more than 19 years old, 87% received two doses of the vaccine. Approximately 80% of the participants received two doses of the measles vaccine, and 10% received only one dose of the vaccine (Table 1). Approximately 4% of participants had a medical history of measles, based on the filled questionnaires.

### 3.2. Measles-IgG Titer and Temporal Characterization

The mean value of the measles-specific IgG antibody titer determined by EIA was 13.4 ± 12.0 (mean ± standard deviation), with values ranging from 0.8 to 128 (Table 2). The mean ages in month for the administration of the first and second doses of the vaccine were 23.6 ± 32.5 and 152.2 ± 25.4, respectively. The recommended age for the administration of only one dose of the measles vaccine was from 12 months to 90 months during 1995 to 2000. Therefore, the duration for the first dose of the vaccine was extended for people born in this period. Almost all participants satisfied the conditions for the special measure recommended for the administration of a second dose of the vaccine at 13 years of age from 2008 to 2013, indicating that more than 80% of participants received two doses of the vaccine during the assigned period. In all vaccinated participants, an average of seven years had passed after the last vaccination. The mean IgG antibody titers of 17- to 18-year-old and 19- to 20-year-old participants were 12.0 ± 12.9 and 11.9 ± 7.8, respectively.

### 3.3. Divergence of Measles-IgG Antibody Titers by Medical History of Measles

To discern the basis of measles IgG antibody titer variation, multivariable regression analysis was performed for the IgG antibody titers and temporal characteristics. It was observed that almost all parameters related to temporal characteristics had multicollinearity. Thus, multivariable regression analysis was performed based on items listed in Table 1, devoid of temporal information. It was found that the distribution of measles IgG antibody titers was significantly correlated with the medical history of measles (*P* < 0.05). The differences in measles IgG antibody titers between individuals with and without a medical history of measles were determined by the unpaired t-tests (Figure 1). The IgG antibody titers collected from participants who had a medical history of measles (27.0 ± 31.8) were significantly higher than titers in participants who had no medical history of measles (12.8 ± 9.9, *P* < 0.05).

Figure 2 shows measles IgG antibody titers in measles vaccinated and unvaccinated participants. The measles IgG antibody titers from the participants who had a medical history of measles and had not received vaccination were excluded from Figure 2 and statistical analysis. The mean IgG antibody titers of participants who received one dose, two doses, and three doses of the measles vaccination exhibited mean values 13.5 ± 16.0, 12.6 ± 8.8, and 12.9 ± 6.7, respectively. Although the mean IgG antibody titer of unvaccinated participants exhibited 13.7 ± 11.6, this cause was still unavailable. The results of the one-way ANOVA showed that there was no significant difference among the IgG titers of individuals based on the number of vaccinations.

### 3.4. Temporal Changes in Measles-IgG Antibody Titers

To verify the temporal changes in measles IgG antibody titers after the last vaccination, the elapsed period was divided into four periods: 12 months or less (*n* = 17, 15.1 ± 10.2), 13–60 months (*n* = 31, 16.0 ± 9.5), 61–72 months (*n* = 279, 12.9 ± 10.6), and 73 months or more (*n* = 73, 10.9 ± 6.8). In the measles vaccine program, almost all participants received two doses of the vaccine at 13 years of age, and, thus, approximately 70% of participants were observed in the period of 61–72 months. Excluding measles IgG antibody titer values from the participants who had a medical history of measles and those who were not vaccinated, one-way ANOVA showed that there were no significant differences among the four periods (Figure 3, *P* = 0.08). The peak measles IgG antibody titer values were intended to be within five years. Thereafter, these titers tend to decrease with time (prolonged period).

Although a gender gap of approximately two folds was present in our study, no significant differences in the IgG antibody titers were observed between female and male participants (*P* ≥ 0.05; Appendix A).

## 4. Discussion

In this study, the participants were comprised of 32% females and 68% males. Several studies have indicated that vaccine-induced IgG antibody titers and the related immune response are different between females and males [10,11]. However, Voigt et al. reported no associations and correlation between biological sex and immunity (humoral or cellular) and measles IgG antibody titers [12]. These results indicate that sex had little influence on the IgG antibody titers in this study.

Protective measles-specific IgG antibody titers determined by EIA corresponded to 4.0, and greater values in a previous study [13], and the seroprevalence of the measles antibody in this study was estimated to be approximately 92%. The seroprevalence is indicated to approach that seen in previous studies that included individuals under a similar vaccination program [14,15]. In order to provide absolute protection from measles virus infection, the measles-specific serum IgG antibody titer (determined by the EIA method) is required to be 12 or more, and in case of an unsatisfactory titer, additional vaccination is recommended [16]. Although the average measles IgG antibody titers of participants overall slightly exceeded 12, measles IgG antibody titers of 56% of the 284 participants showed a value less than 12. These results indicate that more than half of the studied participants require an additional dose of vaccine to protect them against measles. According to a World Health Organization (WHO) announcement, Japan has achieved measles elimination, which was defined as an interruption of endemic measles virus transmission for at least 36 months [17]. Nevertheless, measles cases from foreigners in Japan have occurred sporadically [2,18,19,20,21]. Based on our surveillance, one of the causes for sporadic measles occurrence may be non-persistent protection by measles IgG antibody titers in young adults. When measles was endemic in Japan in the year 2000, approximately 200,000 people, mainly children, were infected with the measles virus [22,23]. Thereafter, an attempt to raise the coverage of the measles vaccination and rigorously enforce the two-dose vaccination program was successfully achieved, which eventually led to the measles elimination period [17]. Currently, although protection against measles infection has been achieved, the young adult population is mainly insufficiently immunized against the measles virus, suggesting that periodic monitoring of the measles epidemic and acquired immunity against the measles virus in young adults is highly required.

This surveillance study focused on the temporal status of measles IgG antibody titers from last vaccination, although there were no significant differences in measles IgG antibody titers after the first dose and second dose of the vaccination. In the United States, the vaccination schedule and patterns are similar to those in Japan. Measles was declared eliminated in the absence of continuous measles transmission for a period greater than 12 months in the United States in the year 2000 [24]. However, endemic outbreaks of measles have rarely been reported in the USA [25,26,27], and the few outbreaks that have occurred were caused in unvaccinated populations [28]. CD (Cluster of differentiation) 46 and TLR (Toll-like receptor) 8 variants were considered to be involved in the one-dose measles vaccine failure [29]. Although it is possible to have such cases, epidemiological studies have demonstrated the efficacy of the measles vaccine. In brief, 95% of children who received the measles vaccine acquired immunity against the measles virus, and two additional doses of vaccine led to more than 99% immunization in children [30,31]. These results revealed that epidemic outbreaks may be caused by the unvaccinated population or a large number of international travellers [4]. According to the large surveillance by healthcare workers, serum measles IgG antibody titers from adults less than 29 years of age showed susceptibility to measles [10]. A single dose of the measles vaccine in adults had significantly increased serum IgG titers, even in the presence of initial insufficient IgG titers [32]. Also, low levels of measles-specific antibody titers have adequate antigen-specific memory B-cell responses [33,34]. Sporadic infection may occur in young and older adults under unprotected conditions, with reduced vaccine-induced IgG antibody titers due to the temporal changes after the last vaccination and the consequent susceptibility to infection. These results also suggest that periodic vaccination against measles is required for young and older adults to prevent the occurrence of even sporadic measles cases.

## Figures and Tables

**Figure 1 vaccines-07-00118-f001:**
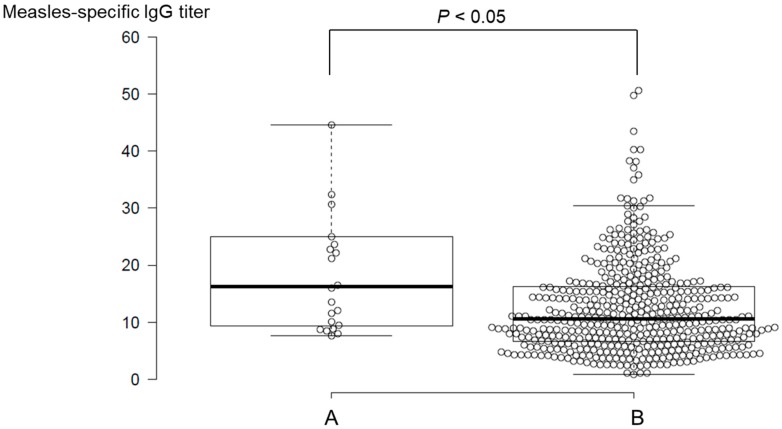
Comparison of measles-specific IgG antibody titers based on the presence (**A**) and absence (**B**) of measles medical history in vaccinated participants. The plots with an IgG antibody titer value of more than 60 are omitted from the figure. The bold lines reveal median values, and the whiskers are extended to data points that are less than 1.5 x interquartile range (IQR) away from 1st/3rd quartile.

**Figure 2 vaccines-07-00118-f002:**
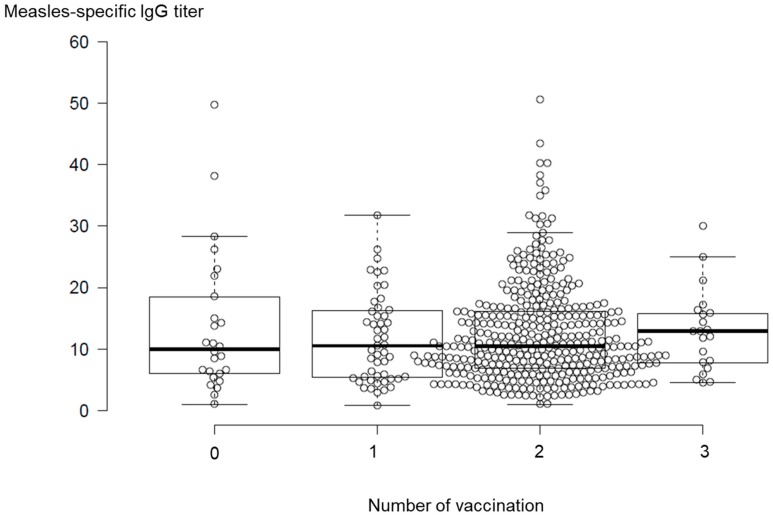
Comparison of measles-specific IgG antibody titers among the number of vaccinations. The plots with an IgG titer value of more than 60 were omitted from the figure. The measles IgG antibody titers from the participants who had measles medical history and unvaccinated participants were excluded from the figure and the statistical analysis. The bold lines reveal median values, and the whiskers are extended to data points that were less than 1.5 x IQR away from 1st/3rd quartile. There were no significant differences in measles-specific IgG antibody titers among vaccinated participants based on one-way ANOVA (*P* = 0.80).

**Figure 3 vaccines-07-00118-f003:**
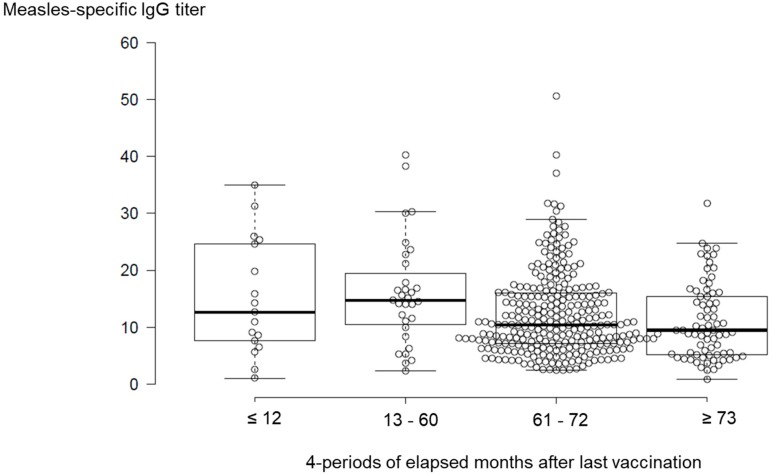
Comparison of measles-specific IgG antibody titers among four elapsed periods after last vaccination (in months). The plots with an IgG titer value more than 60 were omitted from the figure. The measles IgG antibody titers of both the participants who had measles medical history and the unvaccinated participants were excluded from the figure and the statistical analysis. The bold lines reveal median values, and the whiskers are extended to data points that were less than 1.5 x IQR away from 1st/3rd quartile. One-way ANOVA showed that there were no significant differences among the four periods (*P* = 0.08).

**Table 1 vaccines-07-00118-t001:** Details of measles vaccine administered survey participants.

Characteristics	Number	Percentage
Total	506	100
Age, Y		
Range		
17–18	407	80
19–20	97	19
≥21	2	0
Sex		
Female	160	32
Male	346	68
Vaccination		
1	56	11
2	399	79
3	21	4
Unvaccinated	30	6
Medical history of measles	22	4

**Table 2 vaccines-07-00118-t002:** Measles-specific immunoglobulin G (IgG) antibody titer and temporal characteristics of participants.

Item	Mean	Range	Number
Measles-specific IgG titer	13.4	0.8–128	506
1st vaccination month	23.6	11–263	476
2nd vaccination month	152.2	16–265	420
3rd vaccination month	156.0	53–238	21
Elapsed months after last vaccination	83.3	0–237	476

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
