# Peer review of "A Survey of Vaccine-Induced Measles IgG Antibody Titer to Verify Temporal Changes in Response to Measles Vaccination in Young Adults"

_vaccines, 2019, doi:10.3390/vaccines7030118_

Round 1

Reviewer 1 Report

Triggered by the recent outbreaks of measles in Japan among young adults, Sasaki et al try in their manuscript "A survey of vaccine-induced measles IgG antibody titer and the verification of changes in temporal differences of measles vaccination in young adults" to evaluate anti-measles titers in the affected age group depending on their history of measles vaccination or infection. By using a spatially and in terms of age very homogenous patient collective, they describe that measles IgG titers are statistically significantly higher after natural infection than after vaccination or in naive patients. They describe no difference in measles titers between vaccinated or non-vaccinated patients, nor a clear trend of titers when correlating those to the time-point of the last vaccination.

Although of potential interest, this study suffers from a number of points that should be considered:

major points:

1.) The study cohort is highly biased. 2/3 of participants were male, 1/3 female, which is problematic due to potential sex bias, especially in the light of know dramatic sex-specific effects of measles vaccination (Holte et al. 1993). Moreover, the age of the participants seems to be extremely homogenous in terms of age (to be expected of university freshmen) and could be biased for a certain region, since all participants are from a single study sight. Thus, this collective is hardly representative.

2.) It is hardly conceivable, that the population of non-vaccinated probands should have titers statistically not different from vaccinee collectives. This dataset questions the validity of the method of analysis

3.) Time intervals chosen for analysing the effects of time after vaccination on measles titers are extremely inhomogenous: 3 mo - 9 mo - 4 yrs - 1 yr - rest. This seems highly unusual, might bias the result and may be the cause of the observed irregular perturbations of titers 3 mo - 1 yr: titers down; 1 yr - 5 yr: titers up; 5 yr - 6 yr: titers down; 6 yrs - more than 6 yrs: titers constant. These data do not allow to draw any conclusions on time-dependence of antibody titers and are in contrast or at least should be discussed in comparison to previous publications on this topic e.g. Kakoulidou et al. 2013; Kennedy et al. 2019...

4.) Some data sets seem to be contradictory, e.g. 80% of subjects were said to be 18 years of age while 19 or more year old participants had not received a second dose. Nevertheless, 80% of all participants received a second vaccination; this would mean, that 100% of 18 year old participants had received BOTH vaccinations, which seems rather unlikely...

minor points:

language should be checked for grammar and spelling the text displays certain redundancies in language first paragraph of results section seems to be a remnant of the writing phase

Author Response

Reviewer 1

Major changes in revised manuscript

Title has changed to “A survey of vaccine-induced measles IgG antibody titer to verify temporal changes in response to measles vaccination in young adults”. English grammar and technical terms have been checked by the experts in commercially available English proofreading company (The certificate was attached). In former Figure 1, means of graph items (medical history) was unclear. Then, we changed description of item to “A” and “B” and detailed them in figure legend. In former Figure 2, IgG antibody titer of unvaccinated participants was no difference toward to that of 1-3 vaccinated participants. These results were obviously different from actual facts, and we suspected that unvaccinated participants had medical history of measles.  Thus, to avoid misapprehension, we have deleted unvaccinated participants in revised Figure 2. In former Figure 3, the participants were divided into 5-periods of vaccination to attain statistical significance without certain conclusion. Thus, these periods were reexamined, and revised 4-periods were statistical analyzed in revised Figure 3.  These results could not be obtained significant difference; however, elapsed month after last vaccination tended to decrease measles-specific IgG titers.

Response for reviewer 1’s comments

>1.) The study cohort is highly biased. 2/3 of participants were male, 1/3 female, which is problematic due to potential sex bias, especially in the light of know dramatic sex-specific effects of measles vaccination (Holte et al. 1993). Moreover, the age of the participants seems to be extremely homogenous in terms of age (to be expected of university freshmen) and could be biased for a certain region, since all participants are from a single study sight. Thus, this collective is hardly representative.

As reviewer 1 indicated, the participants were composed of 32% female and 68% male. For the comparative study, this component may be biased.  In initial stage of the study, we had argued on this point.

Haralambieva et al. (2014) and Kumakura et al. (2014) reported that vaccine-induced IgG antibody titers and related immune response was different between female and male. However, the reasons underlying differences between female and male are still unknown (both references were cited in revised MS).

For the measles vaccination, Voigt et al. (2016) reported that no associations and correlation between biological sex and either humoral or cellular immunity to measles vaccine was observed in a large-scale study (this study was also cited).

Further, we have confirmed that there were not significant differences on antibody titers between female and male in this study. 

On these points, certain mechanisms are still unavailable. Although we have recognized sex-biased study design, our explanation was newly described in first paragraph of Discussion section (lines 180-187).

>2.) It is hardly conceivable, that the population of non-vaccinated probands should have titers statistically not different from vaccinee collectives. This dataset questions the validity of the method of analysis

As reviewers indicated, statistical analysis concluded that IgG antibody titers in unvaccinated participants were higher than that in participants who had received one or three doses of vaccination in former Figure 2.  These results were caused that several participants with unvaccination might be infected measles, and they did not record and declare own vaccination and medical history.  To include those data in comparison of vaccination effectiveness may be perturbation in this study.  Therefore, we have deleted unvaccinated participants from revised Figure 2 and 3 and compared the participants who had received certain vaccination.

>3.) Time intervals chosen for analysing the effects of time after vaccination on measles titers are extremely inhomogenous: 3 mo - 9 mo - 4 yrs - 1 yr - rest. This seems highly unusual, might bias the result and may be the cause of the observed irregular perturbations of titers 3 mo - 1 yr: titers down; 1 yr - 5 yr: titers up; 5 yr - 6 yr: titers down; 6 yrs - more than 6 yrs: titers constant. These data do not allow to draw any conclusions on time-dependence of antibody titers and are in contrast or at least should be discussed in comparison to previous publications on this topic e.g. Kakoulidou et al. 2013; Kennedy et al. 2019...

In Figure 3, we reconsidered and reconstructed revised Figure 3.  As reviewer 1 indicated,

Interpretation of antibody titers’ distribution might be biased by the period-designation.  Although there were several significant differences by ANOVA and individual t-test, these results could not be led to certain conclusion.  Then, period-designation was simplified even if there are statistically insignificant difference.  In brief, period was divided into 4-terms including “≤12month”, “13-60month”, “61-72month” and “≥73month”.  Thereafter, we have carried out ANOVA again, and these results indicate that there were no significant differences among those periods.  However, P value (0.08) revealed that a tendency was observed in these changes.  Highly titer was observed in “13-60month”, and thereafter, the titers were intended to be decreased.

For terms of “13-60month” and “61-72month”, almost of participants were concentrated 60 and 70 months, respectively.  Therefore, we detailed number of participants and mean±SD values of antibody titer in Results section in lines 137-139.

     Many thanks for kind introduction of article by Kakoulidou and Kennedy et al..  These references were cited in discussion section (lines 224-225).

>4.) Some data sets seem to be contradictory, e.g. 80% of subjects were said to be 18 years of age while 19 or more year old participants had not received a second dose. Nevertheless, 80% of all participants received a second vaccination; this would mean, that 100% of 18 year old participants had received BOTH vaccinations, which seems rather unlikely...

     We would like to correct this paragraph in former MS.  “All the subjects who were 19 years old did not receive second inoculations.” was misconception.  Then, this sentence was deleted in revised MS.

>language should be checked for grammar and spelling the text displays certain redundancies in language first paragraph of results section seems to be a remnant of the writing phase

English proofreading was performed, and English grammar and technical terms was checked and corrected by native speakers.  We would like to attach certification of English proofreading.

Reviewer 2 Report

The first sentence in the Introduction (In Japan, more than 300 cases of measles were reported, which has increased rapidly compared 33 to that in the past 6 years [1].) should specifically indicate the time period of the more than 300 cases. Several sentence at the beginning of Results seems unnecessary.

“This section may be divided by subheadings. It should provide a concise and precise description of the experimental results, their interpretation as well as the experimental conclusions that can be drawn.

3.1. Subsection”

The subsections can be 3.1; 3.2; 3.3; and 3.4.  (No need for 3.1.1; 3.1.2, et al)

Figure 1 compares IgG antibody titers of medical history with no history. It is not clear whether the “no history” includes both the vaccinated and unvaccinated, or only unvaccinated.

Author Response

Reviewer 2

Major changes in revised manuscript

Title has changed to “A survey of vaccine-induced measles IgG antibody titer to verify temporal changes in response to measles vaccination in young adults”. English grammar and technical terms have been checked by the experts in commercially available English proofreading company (The certificate was attached). In former Figure 1, means of graph items (medical history) was unclear. Then, we changed description of item to “A” and “B” and detailed them in figure legend. In former Figure 2, IgG antibody titer of unvaccinated participants was no difference toward to that of 1-3 vaccinated participants. These results were obviously different from actual facts, and we suspected that unvaccinated participants had medical history of measles.  Thus, to avoid misapprehension, we have deleted unvaccinated participants in revised Figure 2. In former Figure 3, the participants were divided into 5-periods of vaccination to attain statistical significance without certain conclusion. Thus, these periods were reexamined, and revised 4-periods were statistical analyzed in revised Figure 3.  These results could not be obtained significant difference; however, elapsed month after last vaccination tended to decrease measles-specific IgG titers.

Response for reviewer 2’s comments

>The first sentence in the Introduction (In Japan, more than 300 cases of measles were reported, which has increased rapidly compared 33 to that in the past 6 years [1].) should specifically indicate the time period of the more than 300 cases.

These incidents were firstly confirmed and reported in May, 2019.  Therefore, detailed period was mentioned as below.  Further, the report that was cited in former MS was changed to detailed version of report published from National Institute of Infectious Diseases in Japan. (lines 37-39).

>Several sentence at the beginning of Results seems unnecessary.

As all the reviewers indicated, this sentence was wasteful.  I missed deletion of this sentence from manuscript form. I would like to apology.

>This section may be divided by subheadings. It should provide a concise and precise description of the experimental results, their interpretation as well as the experimental conclusions that can be drawn.

This sentence was deleted in revised MS.

>3.1. Subsection”

The subsections can be 3.1; 3.2; 3.3; and 3.4.  (No need for 3.1.1; 3.1.2, et al)

Numberings of subsection were corrected as reviewers indicated.

>Figure 1 compares IgG antibody titers of medical history with no history. It is not clear whether the “no history” includes both the vaccinated and unvaccinated, or only unvaccinated.

In Figure 1, IgG antibody titers of vaccinated patients were used.  As reviewers’ suggestions, title of X-axis in Figure 1 was seemed to be unclear.  Title of X-axis was changed to “A” and “B”, and then, initial sentence of Figure 1 legend was changed as blow: Comparison of measles-specific IgG antibody titers based on the medical history (A) and no history (B) of measles in vaccinated participants.   

Reviewer 3 Report

This study examined measles seroprevalence in young adults. Throughout the world – China, Europe, US, and Japan, there is a higher-than-expected burden of measles disease in the young adult population. Seroprevalence studies within these populations can be important to identify high-risk populations and develop efficient control strategies.

Major issues:

What is the interpretation of the titers? as in, is there a cut-off at which point the titer is considered protective? On Table 1, you could provide the mean and sd for the titers in each group, but you could also provide the percent who are considered protected with a given cut-off. You mention using multiple regression, but I don’t think you do – in my mind, multiple regression is synonymous with multivariable regression, where several independent predictors are put into, say, a linear regression model. Which I do not think you did.

Minor issues:

In the abstract, can you mention what city/prefecture/region your study took place? (also put in aim in introduction line 60). line 33 – you mention 300 cases of measles. What year(s) are you referring to? Is record of measles vaccination required for university matriculation? You mention wanting to avoid multicollinearity (line 82), but I’m unaware of how a bivariate analysis of correlation would address this. You could provide more details, or just delete this sentence (with a large-ish sample size like you have, multicollinearity becomes less of a problem) You can delete lines 91-94 – and then probably change 3.1.1. and other sub-sub-sections to a simpler 3.1. subsection. You mention words like “primary” and “secondary” vaccination – I would switch to language of “one dose” or “two doses” of vaccine throughout your paper. Table 1 – change medical history to medical history of measles (the phrase medical history by itself is vague) In Table 1 instead of reporting mean and range for age (or in addition to), you could divide up into several groups, like 17-18, 19-20, etc. and report number and percentage for those. Figures 1-3, could you write out the mean and SD for each of the categories? you do for figure 1 on line 122 but do something similar for other figures. The subject for the sentence on line 121 should be “The mean IgG antibody titer collected from ….” When was measles eliminated in Japan? (mention year on line 167) situate your study in other research. some examples below:

Durrheim D. Measles elimination, immunity, serosurveys, and other immunity gap diagnostic tools. J Infect Dis. 2018;218: 341–343. https://doi.org/10.1093/infdis/jiy138

Thompson KM, Odahowski CL. Systematic Review of Measles and Rubella Serology Studies. Risk Anal. 2015; 1–27. https://doi.org/10.1111/risa.12430

both these papers explain why measles seroprevalence studies are important – and they could be used as background in your introduction for why you conducted your study.

Gohil DJ, Kothari ST, Chaudhari AB, Gunale BK, Kulkarni PS, Deshmukh RA, Chowdhary AS. Seroprevalence of Measles, Mumps, and Rubella Antibodies in College Students in Mumbai, India. Viral Immunol. 2016;29: vim.2015.0070. https://doi.org/10.1089/vim.2015.0070

 Boulton ML, Wang X, Zhang Y, Montgomery JP, Wagner AL, Carlson BF, Ding Y, Li X, Gillespie B, Su X. A population profile of measles susceptibility in Tianjin, China. Vaccine. 2016;34: 3037–3043. https://doi.org/10.1016/j.vaccine.2016.04.094

both these studies examine similar age ranges as your own – a comparison outside of Japan could be useful within your discussion section.

Author Response

Reviewer 3

Major changes in revised manuscript

Title has changed to “A survey of vaccine-induced measles IgG antibody titer to verify temporal changes in response to measles vaccination in young adults”. English grammar and technical terms have been checked by the experts in commercially available English proofreading company (The certificate was attached). In former Figure 1, means of graph items (medical history) was unclear. Then, we changed description of item to “A” and “B” and detailed them in figure legend. In former Figure 2, IgG antibody titer of unvaccinated participants was no difference toward to that of 1-3 vaccinated participants. These results were obviously different from actual facts, and we suspected that unvaccinated participants had medical history of measles.  Thus, to avoid misapprehension, we have deleted unvaccinated participants in revised Figure 2. In former Figure 3, the participants were divided into 5-periods of vaccination to attain statistical significance without certain conclusion. Thus, these periods were reexamined, and revised 4-periods were statistical analyzed in revised Figure 3.  These results could not be obtained significant difference; however, elapsed month after last vaccination tended to decrease measles-specific IgG titers.

>What is the interpretation of the titers? as in, is there a cut-off at which point the titer is considered protective? On Table 1, you could provide the mean and sd for the titers in each group, but you could also provide the percent who are considered protected with a given cut-off.

As indicated, we avoided make a reference of seroprevalence and protective titers since the value of EIA varied according to certain conditions in former MS.  However, general seroprevalence and mean±SD in this study was mentioned in Discussion (lines 188-196) and Results (lines 116-117), respectively.

>You mention using multiple regression, but I don’t think you do – in my mind, multiple regression is synonymous with multivariable regression, where several independent predictors are put into, say, a linear regression model. Which I do not think you did.

We used multivariable regression in this study.  Then, description of “multiple regression” was corrected to “multivariable regression”.

>In the abstract, can you mention what city/prefecture/region your study took place? (also put in aim in introduction line 60).

Prefecture name was added in abstract, introduction and methods section as below.

Abstract: lines 20-22.

Introduction: lines 71-68.

Materials and methods: lines 74-81.

>line 33 – you mention 300 cases of measles. What year(s) are you referring to?

These incidents were firstly confirmed and reported in May, 2019.  Therefore, detailed period was mentioned as below.  Further, the report that was cited in former MS (reference #2) was changed to detailed version of report published from National Institute of Infectious Diseases in Japan. These descriptions were added into lines 37-39 in Introduction section.

>Is record of measles vaccination required for university matriculation?

In Japan, almost universities do not require vaccination record, and then measles prevails in young adults mainly university students.

>You mention wanting to avoid multicollinearity (line 82), but I’m unaware of how a bivariate analysis of correlation would address this. You could provide more details, or just delete this sentence (with a large-ish sample size like you have, multicollinearity becomes less of a problem)

In former MS (line 82): To avoid multicollinearity, factors were preliminary determined with bivariate analysis and were confirmed to have no correlation.

As reviewers indicated, this sentence was interpreted as a meaningless.  Then, we have deleted in revised MS.

>You can delete lines 91-94 – and then probably change 3.1.1. and other sub-sub-sections to a simpler 3.1. subsection.

The unnecessary sentence that was used for paper preparation was deleted in revised MS.

Numberings of subsection were corrected as reviewers indicated.

>You mention words like “primary” and “secondary” vaccination – I would switch to language of “one dose” or “two doses” of vaccine throughout your paper.

In revised MS, terms of primary and secondary vaccination were changed to those of one dose and two doses of vaccine, respectively.  Further, “first dose” or “second dose” was used if the sentence means experience of vaccination.

>Table 1 – change medical history to medical history of measles (the phrase medical history by itself is vague)

In Table 1, “Medical history” has changed to “Medical history of measles”.

According to this change, similar description in Figures and main text in anywhere has been corrected as mentioned below in revised MS.

In Figure 1; Also, the title of X-axis was same a word of ambiguous meaning in Figure 1 of former MS, and therefore, that title was simply changed to “A” and “B”.  Due to this change, Figure 1 legend was changed to “Comparison of measles-specific IgG antibody titers based on the medical history (A) and no history (B) of measles in vaccinated participants.” (lines 148-149)

In Figure 2 and 3legend; lines 156-157 and 165-166.

In abstract section; lines 24-26.

In results 3-3 subheading; line 118.

In results 3-3 subsection; lines 126-128.

In results 3-3 subsection; lines 129-131.

In results 3-4 subsection; lines 142-144.

>In Table 1 instead of reporting mean and range for age (or in addition to), you could divide up into several groups, like 17-18, 19-20, etc. and report number and percentage for those.

In Table1, description of age has been changed according to the reviewer’s suggestion.

In brief, age range was divided into 3 ranges such as 17-18, 19-20 and more than 21.  Number of 21-24 year was very small, and therefore, those ranges was bundled as “≥ 21”.

>Figures 1-3, could you write out the mean and SD for each of the categories? you do for figure 1 on line 122 but do something similar for other figures.

As for Figure 2 and 3, X-axis of vaccination numbers and time-periods were changed, respectively. All the corrected values of mean±SD were described in text section (Figure 1; lines 126-128, Figure 2; 131-135, Figure 3; 137-139).

> When was measles eliminated in Japan? (mention year on line 167) situate your study in other research. some examples below:

In Introduction, we have newly added measles elimination month (lines 35-37).

>both these papers explain why measles seroprevalence studies are important – and they could be used as background in your introduction for why you conducted your study.

Many thanks for valuable information.  We would cite these 2 articles (Durrheim, 2018; Thompson et al., 2015) in Introduction section (lines 63-65).

>both these studies examine similar age ranges as your own – a comparison outside of Japan could be useful within your discussion section.

These 2 articles (Gohil et al., 2016; Boulton et al., 2016) were newly cited in Discussion section (lines 190-191).

Round 2

Reviewer 1 Report

The authors of the study have aimed to resolve the points of critique, but have not been able to convince the reviewer:

Simply excluding the incovenient data set of the unvaccinated control group on the basis of pure speculation that the control group must have had the disease without any further evidence is not the solution for the underlying problem. 

Reporting that there may be a gender bias in the study does not resolve the problem of the bias, itself. The data set must at least be stratified or balanced, in the reviewer´s opinion, to draw any valid conclusions. Moreover, the author´s do not comment on the regional aspect of data bias. Therefore, general conclusions can hardly be drawn.

The unusual intervals for analysis presented in Fig. 3 has not been resolved by merging two cohorts. The intervals still are 12 mo - 48 mo - 12 mo - rest, which is neither uniformly distributed nor reflecting any other obvious dsitribution.

The reviewer is furthermore surprised by the supposed misconception, that no subject exceeding 19 years of age received a second shot. This is not convincing without revealing actual numbers, the authors should be able to provide.

Author Response

Major revision

We have changed form of 3 figures to “bee swarm” in order to understand distribution and prevalence of actual number in graph column. 

Response to comments

>Simply excluding the incovenient data set of the unvaccinated control group on the basis of pure speculation that the control group must have had the disease without any further evidence is not the solution for the underlying problem.

We have incorporated unvaccinated participants again in revised Figure 2.  However, we could not find out why the mean titer and median value of unvaccinated population exhibited similar to vaccinated population as reviewer indicated.  Then, we added sentence that the cause is unavailable in Results section (lines 135-136).

>Reporting that there may be a gender bias in the study does not resolve the problem of the bias, itself. The data set must at least be stratified or balanced, in the reviewer´s opinion, to draw any valid conclusions. Moreover, the author´s do not comment on the regional aspect of data bias. Therefore, general conclusions can hardly be drawn.

     We have newly generated supplementary figures that compared measles-specific IgG antibody titers sorted by male and female (Supplementary figure 1-3).  We have compared IgG antibody titers by t-test and have confirmed no differences between the sexes.  Thus, the sentence that there were no differences has been added in end of Results section (lines 149-151).  As reviewer indicated, however, it is well-known that there is difference in gender of measles-specific antibody titer in several study, and therefore, the first paragraph has been left as previously (189-193).  We would like to provide recent articles for the readers regarding the gender-bias even if it may be regional aspect.

>The unusual intervals for analysis presented in Fig. 3 has not been resolved by merging two cohorts. The intervals still are 12 mo - 48 mo - 12 mo - rest, which is neither uniformly distributed nor reflecting any other obvious dsitribution.

     The WHO recommends two doses of vaccine such as 12 month of age, 6 and 12 years old.  In Japan, many people are received 2 doses of measles vaccine excluding the exception. 

The individual periods are considered to be too bias to allocate uniformly in Figure 3.  The intervals seemed to be unevenness; however, allocation is thought to be adequate for graph designation.  Then, we would provide graph that described distribution of elapsed months after last vaccination for the reviewer.  Please find below graph.

>The reviewer is furthermore surprised by the supposed misconception, that no subject exceeding 19 years of age received a second shot. This is not convincing without revealing actual numbers, the authors should be able to provide.

We had failed the description of this content.  As recalculated number of vaccinations, 87% of the participants who are more than 19 years old received two doses of measles vaccine.  We would correct the description in former MS and add above sentence (lines 103-104).
